# Effect of switching from nucleos(t)ide maintenance therapy to PegIFN alfa-2a in patients with HBeAg-positive chronic hepatitis B: A randomized trial

Hyun Young Woo[ID][1], Jeong Heo[ID][1]*, Won Young Tak[2]*, Heon Ju Lee[3], Woo Jin Chung[4], Jung Gil Park[3], Soo Young Park[2], Young Joo Park[1], Yu Rim Lee[2], Jae Seok Hwang[4], Young Oh Kweon[2]

1 Department of Internal Medicine, College of Medicine, Pusan National University and Medical Research Institute, Pusan National University Hospital, Busan, Republic of Korea, 2 Department of Internal Medicine, College of Medicine, Kyungpook National University, Kyungpook National University Hospital, Daegu, Republic of Korea, 3 Department of Internal Medicine, Yeungnam University College of Medicine, Daegu, Republic of Korea, 4 Department of Internal Medicine, Keimyung University School of Medicine, Daegu, Republic of Korea

* jheo@pusan.ac.kr (JH); wytak@knu.ac.kr (WYT)

**Data Availability Statement:** All relevant data are within the paper and its Supporting Information files.

## Abstract

### Aims

Induction of a durable viral response is difficult to achieve in patients with chronic hepatitis B (CHB), even from long-term use of a nucleos(t)ide analogue (NA). This study investigated whether switching to peginterferon (PegIFN) alfa-2a after long-term NA therapy induced a durable viral response.

### Methods

Patients with hepatitis B e antigen (HBeAg)-positive CHB who received any NA for at least 72 weeks and had a low level of HBV DNA ($\leq$100 IU/mL) were randomized (1:1) to receive PegIFN alfa-2a (180 μg/week) or NA for 48 weeks. The primary endpoint was change in the hepatitis B surface antigen (HBsAg) titer during antiviral therapy.

### Results

We randomized 149 CHB patients to the two groups. Compared to baseline, the HBsAg levels in both groups were not lower at week 12, but were lower after 24, 36, and 48 weeks (all p<0.001). The maximal HBsAg decline in the PegIFN alfa-2a group was at week 36 (0.50 ±0.88 log$_{10}$ IU/mL), and this decline was smaller in the NA group (0.08±0.46 log$_{10}$ IU/mL). The percentage of patients with HBeAg seroconversion at week 48 was also greater in the PegIFN alfa-2a group (15/75 [20.0%] *vs.* 5/74 [6.8%], p = 0.018). Multivariable analysis indicated the PegIFN alfa-2a group had a greater change in HBeAg seroconversion at week 48 (p = 0.027). Patients had relatively good tolerance to PegIFN alfa-2a therapy.

**Funding:** JH was supported by an unrestricted grant from Roche Korea and by the National Research Foundation of Korea (NRF) grant funded by the Korea government (MSIT) (No. 2021R1F1A1052110). Roche Korea had no role in the study design data collection and analysis, decision to publish, or preparation of the manuscript and the authors retain full responsibility for the collection and interpretation of data, decision to publish, and preparation of the manuscript. https://www.roche.co.kr/ https://www.nrf.re.kr/eng/main/.

**Competing interests:** I have read the journal's policy and the authors of this manuscript have the following competing interests: [JH received a grant from Roche Korea, HYW, WYT, HJL, WJC, JGP, YJP, YRL, SYP, YOK and JSH report no conflicts of interest.]. This does not alter our adherence to PLOS ONE policies on sharing data and materials.

## Conclusions

CHB patients who switched to PegIFN alfa-2a for 48 weeks had a significantly lower HBsAg titer and increased HBeAg seroconversion relative to those who remained on NA therapy.

## Trial registration

(ClinicalTrials.gov; NCT01769833).

## Introduction

The use of potent oral nucleos(t)ide analogues (NAs) to treat patients with chronic hepatitis B (CHB) can reduce the rate of hepatitis B virus (HBV) -associated disease progression, including the development of hepatocellular carcinoma and hepatic decompensation, and dramatically improve patient survival [1–4]. However, even after long-term NA treatment of these patients, many of them do not achieve a durable viral response (i.e., a sustained response following NA cessation), such as seroclearance of HB surface antigen (HBsAg), seroconversion of HB e-antigen (HBeAg) with undetectable HBV DNA. Instead, long-term NA treatment can increase the risk of viral resistance. Functional cure, defined as undetectable HBV DNA and sustained loss of detectable HBsAg [5], is associated with favorable clinical outcome, and these patients can discontinue NA treatment without experiencing virological relapse [6–8].

Other research reported that treatment with peginterferon (PegIFN) alfa-2a for 1 year led to a higher rate of HBsAg loss (3–7%), and this increased to 12% at 5 years after discontinuation [9–11]. HBeAg seroconversion is also sustained after discontinuation of PegIFN alfa-2a treatment, reaching 36% at 6 months and 83% at 1 year [12–14]. However, PegIFN alfa-2a is not effective in all patients, and is associated with higher rates of adverse events [15]. Thus, most CHB patients are currently treated with oral NAs rather than PegIFN alfa-2a.

Although efforts are underway to develop novel therapies that can achieve a functional cure of CHB [8, 16], a durable viral response remains difficult to achieve and sustain. Several trials have tested the ability of PegIFN alfa-2a to provide HBeAg seroconversion in HBeAg-positive patients following long term NA treatment. Two studies showed that the concurrent use of lamivudine plus PegIFN alfa-2a did not provide greater efficacy than PegIFN alfa-2a or oral NA alone [12, 17]. In contrast, another study found that the rate of HBeAg seroconversion was higher in patients who switched from entecavir to PegIFN alfa-2a than in those who continued NA monotherapy [18]; however, patients in this previous study were highly selected from a population of patients who were most likely respond to PegIFN alfa-2a therapy.

The objective of the present study was to compare the effects of a 48-week treatment with PegIFN alfa-2a or NA on HBsAg change and HBeAg seroconversion in NA-controlled HBeAg-positive CHB patients.

## Methods

### Study population

The study protocol was approved by the independent ethics committee of each participating institution, and all included patients provided written informed consent. This study was registered at ClinicalTrials.gov as NCT01769833. This phase III, randomized, open-label trial of CHB patients compared the effect of switching to PegIFN alfa-2a after long-term NA therapy with the effect of continuing NA therapy on the level of HBsAg and HBeAg seroconversion.

Patients in the two groups were recruited from four Liver Centers in South Korea between August 2013 and December 2020. Patients were eligible if they were HBeAg-positive before NA treatment, received any NA-regimen except telbivudine for at least 18 months, had an HBV DNA level of 100 IU/mL or less for at least 12 months (the detection limit at study onset in 2012), and had an HBsAg titer of 100 IU/mL or more at enrollment. The other inclusion criteria were male or female over the age of 20 years; alanine aminotransferase (ALT) level less than 10-times the upper limit of normal (ULN); negative urine or serum pregnancy test result within 24 hours before the first administration of the test drug (for women of childbearing age); ability to participate in the treatment and follow-up observations and to follow the research protocol; and receipt of written informed consent from the participant. The exclusion criteria were presence of decompensated cirrhosis (Child score B–C); history of spontaneous bacterial peritonitis, bleeding due to varicose veins, hepatic encephalopathy, or other signs of liver dysfunction; clinical or radiographic evidence suggesting hepatocellular carcinoma; concomitant infection with the hepatitis C virus (HCV) or HIV; other causes of liver disease; pregnancy or lactating; use of an immunomodulatory or immunosuppressant agent within 6 months prior to registration; serious additional disease that could affect the test results (e.g., congestive heart failure, kidney failure, chronic pancreatitis, uncontrolled diabetes, alcoholism, malignant tumors, etc.); receipt of or planning for a liver transplant; history of hypersensitivity to interferon; resistance to the NAs currently being administered; or previous use of telbivudine.

At the beginning of the study, there was a one-to-one random assignment of patients to receive 180 μg/week PegIFN alfa-2a or continue daily oral NA therapy for 48 weeks. A stratified probability scheme was applied for drug allocation and patients were allocated according to a randomization table separately prepared for each participating hospital. To reduce the risk of ALT flares in patients switching to PegIFN alfa-2a, patients randomized to this group continued taking oral NAs for the first 12 weeks of PegIFN alfa-2a treatment. Patients in the PegIFN alfa-2a group who experienced HBV DNA elevation (viral breakthrough), with or without ALT flare, were allowed to restart concomitant oral NA at the discretion of the researchers. In case of viral breakthrough in the NA group, rescue management, such as adding an NA or changing the NA, was performed according to contemporary HBV guidelines based on the mutation profile and patient compliance, and at the discretion of the researchers. Viral breakthrough in the PegIFN alfa-2a group was defined as an increase in HBV DNA to 2000 IU/mL or more during treatment, and viral breakthrough in the NA group was defined as an increase in HBV DNA (IU/mL) by 10-fold or more from its nadir on more than one occasion during NA treatment. PegIFN alfa-2a was not discontinued in case of viral breakthrough. The criteria for dropout were severe adverse effects or withdrawal of consent to participate.

This study was performed in accordance with Good Clinical Practices and the ethical principles of the Declaration of Helsinki.

## Efficacy

The primary endpoint was the change in HBsAg quantity ($\log_{10}$ HBsAg) during drug administration in each group. The secondary endpoints were (*i*) changes in serum HBV DNA level from baseline and the ratio of below 20 IU/mL (the detection limit), below 2000 IU/mL, and below 20,000 IU/mL during drug administration in each group and (*ii*) change of HBeAg seroconversion and loss during drug administration in each group.

After randomization, patients in each group were evaluated at baseline and every 12 weeks until the end of treatment (week 48). Each visit included measurements of vital signs, a

physical examination, routine laboratory measurements (complete blood count, liver bio-chemistry, hepatitis B serology, serum HBV DNA, and quantitative HBsAg titer), and determi-nation of adverse events. An interim analysis was conducted 24 weeks after trial initiation. Serum concentrations of HBsAg, anti-HBs antibody, HBeAg, and anti-HBe antibody were measured every 3 months using the ARCHITECT i2000SR immunoassay analyzer (Abbott, Illinois, U.S.A.). Routine biochemical tests (serum concentrations of ALT, aspartate amino-transferase [AST], albumin, total bilirubin, and creatinine) were performed using a sequential multiple autoanalyzer. HBV DNA was measured using a real-time PCR assay on a Cobas Taq-Man 48 Analyzer (Roche Molecular Diagnostics, Branchburg, NJ, USA), and had a detection limit of 20 IU/mL. HBsAg concentration was quantified using the Elecsys® HBsAg II Quant Assay (Roche Diagnostics, Indianapolis, IN, USA). Adverse events were classified according to the Medical Dictionary for Drug Regulatory Affairs using World Health Organization termi-nology. A severe adverse event was defined as an event that led to study drug discontinuation.

## Statistical analysis

Previous results suggested that the mean ± standard deviation (SD) between-group difference in the change of HBsAg titer would be 0.6 (±1.1) $\log_{10}$ IU/mL [19]. Thus, 72 patients per treat-ment group would have a statistical power of at least 90% for detecting a between-group differ-ence of 0.6 (±1.1) $\log_{10}$ IU/mL at a significance level of 0.05. Assuming a dropout rate of 10%, the sample size was 80 patients per group. The primary end points of the two groups were compared using an independent samples $t$-test.

Data are presented as numbers with percentages (categorical variables) or means ± SDs (continuous variables). Differences in patient characteristics were compared using the chi-square test or Fisher's exact test (categorical variables) or the independent samples $t$-test or Mann-Whitney's U test (continuous variables). The Shapiro-Wilk test was used to determine the normality of distributions.

Considering the unbalanced nature of the repeated measurement data, a linear mixed model (LMM) with random intercepts was used for statistical fitting. The LMM model included repeated measures of continuous variables as dependent variables; group, time, and group×time interaction as fixed effects; baseline outcome as a continuous covariate; and study participant as a random effect. To avoid making assumptions about the covariance structure, an unstructured covariance matrix that was allowed to differ between groups for the LMM analysis was used, and the Bonferroni procedure was applied in post-hoc analyses. A general-ized estimating equation (GEE) model was used to examine changes in seroconversion from baseline in the two groups. The response variable (seroconversion) was a binary variable (1 if yes and 0 otherwise); the fixed effects were time, group, and time×group interaction; and study participant was a random effect. The model accounted for within-subject correlation in out-comes using the appropriate covariance structure, as determined by the quasi-information cri-teria (QIC).

Univariate and multivariate logistic regression analyses were performed to identify prog-nostic factors that were independently related to seroconversion.

Intention to treat (ITT) analysis was performed to evaluate treatment safety and efficacy. For the continuous variables included in the efficacy evaluation, the last observation carried forward (LOCF) method was used to impute missing values related to the treatment period. In the evaluation of the binary efficacy endpoints, missing values were considered as failures; for subjects who restarted concomitant oral NA due to HBV DNA elevation (viral breakthrough), the same rule was applied. Sensitivity analyses were used to assess the robustness of the find-ings from the primary analysis. In particular, post-hoc sensitivity analyses on the primary

endpoint were provided, with presentation of the corresponding results. All statistical analyses were carried out using SPSS version 26.0, and a *p* value less than 0.05 was considered statistically significant.

## Results

### Baseline characteristics of patients

We screened 174 patients for eligibility, and randomized 149 patients, 75 to the PegIFN alfa-2a group and 74 to the NA group (Table 1). Overall, the mean age was 46.26 ± 10.68 years, 75.8% of the patients were male, and the two groups had similar clinical characteristics at baseline. The most common antiviral agent in both groups was entecavir, followed by lamivudine plus adefovir. Seven patients in the PegIFN alfa-2a group and 5 in the NA group dropped out before week 48; five patients the PegIFN alfa-2a group dropped out due to adverse events considered to be related to this drug (Fig 1).

### Change of HBsAg titer during 48 weeks of treatment

Baseline measurements (Fig 2A and S1 Table) indicated the mean $\log_{10}$ HBsAg level was similar in the PegIFN alfa-2a group (3.50±0.55 $\log_{10}$ IU/mL) and the NA group (3.49±0.51 $\log_{10}$ IU/mL). At week 12, these levels were not different from baseline (p = 0.103), but they were lower than baseline in both groups after week 24 (all p < 0.001). Notably, the amount of HBsAg decline was different between the groups (p = 0.033), and the interaction between group and time was significant (p = 0.033; S1 Table). Post-hoc analyses indicated a significant decline of HBsAg after week 24 in the PegIFN$\alpha$-2a group, and a statistically significant but smaller decrease in the NA group at these times. These results

**Table 1. Baseline characteristics of patients in the two groups\*.**

| Variable | Overall (n = 149) | Group PegIFNα-2a (n = 75) | NA (n = 74) |
|---|---|---|---|
| Age (years) | 46.26±10.68 | 45.29±9.39 | 47.24±11.82 |
| Gender | | | |
| Male | 113 (75.8%) | 60 (80.0%) | 53 (71.6%) |
| Body mass index, kg/m$^2$ | 23.82±3.24 | 23.67±3.28 | 24.04±3.20 |
| Liver cirrhosis | 17 (11.4%) | 6 (8.0%) | 11 (14.9%) |
| Duration of previous NA treatment (months) | 72.89±40.16 | 77.36±45.48 | 68.37±33.64 |
| HBsAg (log$_{10}$IU/mL) | 3.50±0.53 | 3.50±0.55 | 3.49±0.51 |
| HBV DNA (log$_{10}$IU/mL) | 0.07±0.34 | 0.06±0.32 | 0.09±0.37 |
| AST (U/L) | 24.67±8.68 | 23.85±8.38 | 25.49±8.95 |
| ALT (U/L) | 26.97±20.66 | 26.12±22.60 | 27.82±18.63 |
| Previous antiviral agents | | | |
| Entecavir | 66 (44.3%) | 28 (37.3%) | 38 (51.4%) |
| Lamivudine | 13 (8.7%) | 4 (5.3%) | 9 (12.2%) |
| Tenofovir | 9 (6.0%) | 2 (2.7%) | 7 (9.5%) |
| Lamivudine + Adefovir | 20 (13.4%) | 12 (16.0%) | 8 (10.8%) |
| Entecavir + Adefovir | 16 (10.7%) | 14 (18.7%) | 2 (2.7%) |
| Entecavir + Tenofovir | 14 (9.4%) | 11 (14.7%) | 3 (4.1%) |
| Lamivudine + Tenofovir | 11 (7.4%) | 4 (5.3%) | 7 (4.1%) |

\*Data are presented as mean±SD or number (%).

NA, nucleos(t)ide analogues; PegIFNα-2a, peginterferon α-2a; HBsAg, Hepatis B surface Antigen; AST, aspartate transaminase; ALT, alanine transaminase.

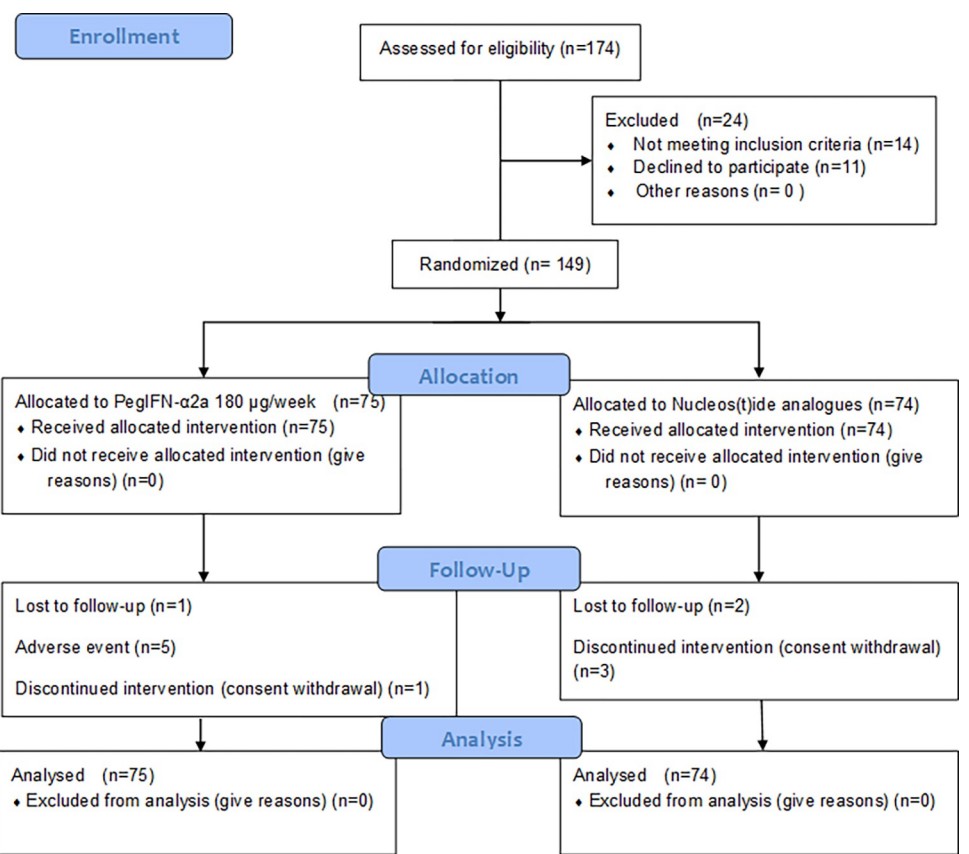

**Fig 1. CONSORT flow diagram.** PegIFN alfa-2a, peginterferon alfa-2a.

indicated that the decline of the HBsAg titer was much greater in the PegIFN alfa-2a group than in the NA group (Fig 2B).

In comparing PegIFNα-2a with NA, we also performed an ITT analysis using LOCF for missing data, ITT analysis without LOCF, and a complete case analysis ignoring the missing data (S2 Table). A complete case analysis is a less conservative approach than ITT analysis. These results showed the effects of PegIFNα-2a on HBsAg were the same in all three analyses.

## Serologic response during 48 weeks of treatment

The percentage of patients with HBeAg seroconversion at week 48 was higher in the PegIFN alfa-2a group (15/75 [20.0%] *vs.* 5/74, [6.8%], p = 0.018; Fig 2C). The percentage of patients with seroconversion also increased over time (p < 0.001) and differed between the two groups (p = 0.015, S1 Table). Post-hoc analyses indicated that the significant improvement in seroconversion was related to PegIFNα-2a treatment. The percentage of patients with seroconversion was significantly greater at week 36 week than at week 12 in the PegIFNα-2a group, but not in the NA group.

There were increases in the percentages of patients with loss of HBeAg during the four assessment times (all p < 0.001), but there were no differences between the two groups (p = 0.283; S1 Table and Fig 2D). One patient in the PegIFN alfa-2a group had HBsAg at loss week 48, but no patients in the NA group had HBsAg loss.

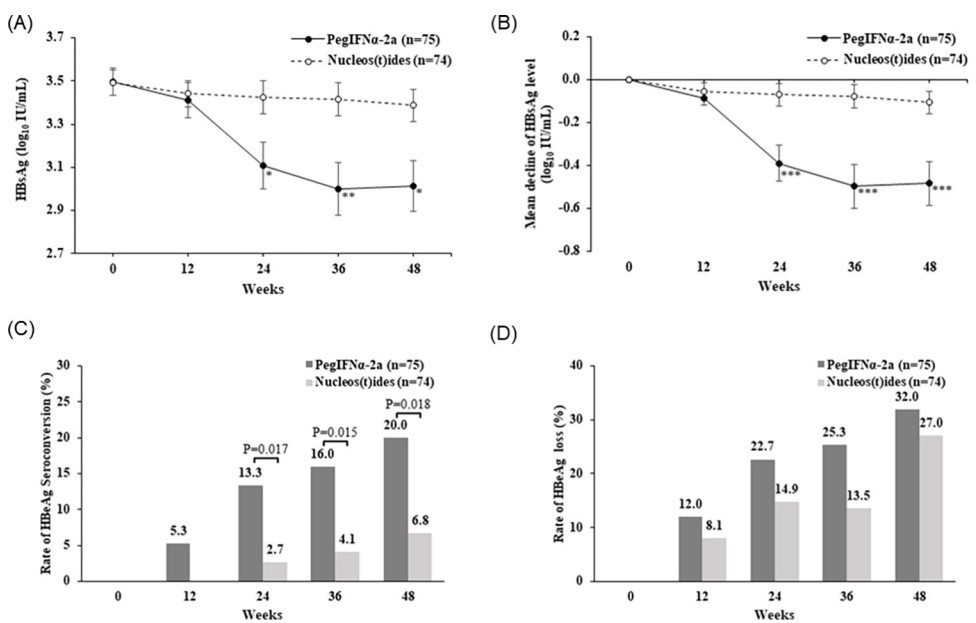

**Fig 2.** Changes of HBsAg level (A), HBsAg reduction (B), HBeAg seroconversion (C) and HBeAg loss (D) during treatment between groups. HBsAg, hepatitis B surface antigen; HBeAg, hepatitis B e Antigen; PegIFN alfa-2a, peginterferon alfa-2a. * p < .05, ** p < .01, *** p < .001.

## HBV DNA elevation during 48 weeks of treatment

Baseline measurements (Fig 3 and S3 Table) indicated the mean HBV DNA level was similar in the PegIFN alfa-2a group ($0.06\pm0.32$ $\log_{10}$ IU/mL) and the NA group ($0.09\pm0.37\log_{10}$ IU/mL). From week 12, HBV DNA level was much greater in the PegIFN alfa-2a group than in the NA group (all p < 0.001). During the 48 weeks of treatment, the percentage of patients in

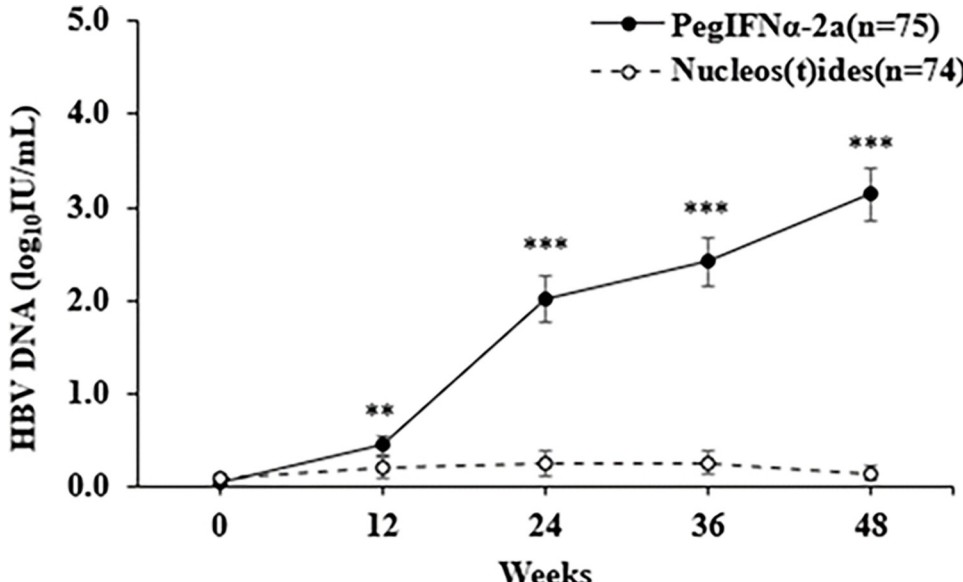

**Fig 3. Changes in serum HBV DNA levels in the two groups.** PegIFN alfa-2a, peginterferon alfa-2a. ** p < .01, *** p < .001.

**Table 2. HBV DNA elevation during 48 weeks of treatment.**

| Variable | Group PegIFNα-2a (n = 75) * | NA (n = 74) * | p | Group PegIFNα-2a (n = 75) ¶ | NA (n = 74) ¶ | p |
|---|---|---|---|---|---|---|
| HBV DNA≥2,000 IU/mL | | | | | | |
| baseline | 0 (0.0) | 0 (0.0) | 1.000 | 0 (0.0) | 0 (0.0) | 1.000 |
| 12 weeks | 1 (1.3) | 1 (1.4) | 1.000 | 1 (1.3) | 1 (1.4) | 1.000 |
| 24 weeks | 18 (24.0) | 0 (0.0) | <0.001 | 18 (24.0) | 0 (0.0) | <0.001 |
| 36 weeks | 9 (12.0) | 0 (0.0) | 0.003 | 21 (28.0) | 0 (0.0) | <0.001 |
| 48 weeks | 6 (8.0) | 0 (0.0) | 0.028 | 24 (32.0) | 0 (0.0) | <0.001 |
| HBV DNA≥20 IU/mL | | | | | | |
| baseline | 3 (4.0) | 4 (5.4) | 0.719 | 3 (4.0) | 4 (5.4) | 0.719 |
| 12 weeks | 13 (17.3) | 3 (4.1) | 0.026 | 15 (20.0) | 4 (5.4) | 0.012 |
| 24 weeks | 30 (40.0) | 1 (1.4) | <0.001 | 38 (50.6) | 4 (5.4) | <0.001 |
| 36 weeks | 10 (13.3) | 3 (4.1) | 0.045 | 38 (50.6) | 5 (6.7) | <0.001 |
| 48 weeks | 6 (8.0) | 1 (1.4) | 0.116 | 42 (56.0) | 3 (4.1) | <0.001 |
| HBV DNA≥20,000 IU/mL | | | | | | |
| baseline | 0 (0.0) | 0 (0.0) | 1.000 | 0 (0.0) | 0 (0.0) | 1.000 |
| 12 weeks | 0 (0.0) | 1 (1.4) | 0.497 | 0 (0.0) | 1 (1.4) | 1.000 |
| 24 weeks | 13 (17.3) | 0 (0.0) | <0.001 | 13 (17.3) | 0 (0.0) | 0.001 |
| 36 weeks | 3 (4.0) | 0 (0.0) | 0.245 | 11 (14.7) | 0 (0.0) | 0.001 |
| 48 weeks | 10 (13.3) | 0 (0.0) | 0.001 | 19 (25.3) | 0 (0.0) | <0.001 |

*Values indicate only patients who newly developed HBV DNA elevation at each time point; patients who developed HBV DNA elevation at a previous time point are not included.

¶ Values indicate the proportion of patients with HBV DNA level >20 IU/mL, 2000 IU/mL and 20,000 IU/mL at each time point, excluding cases with follow-up loss or with rescue NA retreatment.

NA, nucleos(t)ide analogues; PegIFNα-2a, peginterferon α-2a.

the PegIFN alfa-2a group who newly developed an HBV DNA level of 2000 IU/mL or more was 1.3% at week 12, 24% at week 24, 12% at week 36, and 8% at week 48 (Table 2). The percentage of patients in the NA group with an HBV DNA level of 2000 IU/mL or more was 1.4% at week 12, and 0% at all other times. Analysis of HBV DNA levels of 20 IU/mL or more and of 20,000 IU/mL or more indicated similar between-group differences (Table 2).

In addition, among patients in the PegIFN alfa-2a group who experienced HBV DNA elevation (mostly above 2000 IU/mL), 1.3% started concomitant NA at week 12, 9.3% started concomitant NA at week 24, 4% started concomitant NA at week 36, and 32% started concomitant NA at week 48 (Table 3).

**Table 3. Retreatment with nucleos(t)ides analogues in the PegIFN alfa-2a group.**

| NA retreatment | PegIFNα-2a (n = 75)† | Incidence* | 95% CI |
|---|---|---|---|
| baseline | 0 (0.0%) | 0.0% | 0.0–0.0% |
| 12 weeks | 1 (1.3%) | 1.3% | 0.0–3.8% |
| 24 weeks | 7 (9.3%) | 10.7% | 3.6–17.8% |
| 36 weeks | 3 (4.0%) | 14.7% | 6.7–22.7% |
| 48 weeks | 24 (32.0%) | 46.7% | 35.3–58.1% |

† Values indicate noncumulative numbers and percentages.

*Cumulative percentage.

NA, nucleos(t)ide analogues; PegIFNα-2a, peginterferon α-2a; CI, confidence interval.

## ALT level during 48 weeks of treatment

Analysis of adverse events indicated 77 patients experienced 86 events of ALT elevation (62 in the PegIFN alfa-2a group and 24 in the NA group) during the 48 weeks of treatment (Table 4). Among these 86 events, 15 in the PegIFN alfa-2a group and 1 in the NA group appeared to be associated with viral breakthrough, based on the transient HBV DNA elevation that followed ALT elevation. Analysis of these 16 incidents indicated the ALT elevation in the PegIFN alfa-2a group was 1–5×ULN (12 incidents), 5–10×ULN (2 incidents), and more than 10×ULN (1 incident); the one incident in the NA group was an elevation of more than 10×ULN. Forty-seven incidents of ALT elevation in the PegIFN alfa-2a group occurred within 3 months after beginning PegIFN alfa-2a in patients who had sustained HBV DNA suppression due to 3 months of concomitant NA treatment. Analysis of these 47 incidents indicated the ALT elevation was 1–5×ULN (45 incidents) and 5–10×ULN (2 incidents). Twenty-three incidents of ALT elevation in the NA group were not associated with virologic rebound or breakthrough, and the elevation in these 23 incidents was 1–5×ULN.

## Safety

The rates of adverse events were significantly higher in the PegIFN alfa-2a group than in the NA group but the rate of serious adverse events were similar between the two groups (Table 4). Dose modification and discontinuation due to safety issues during treatment only occurred in the PegIFN alfa-2a group. Ten patients in the PegIFN alfa-2a group discontinued treatment for safety reasons; 5 patients permanently discontinued this treatment due to

**Table 4.  Rates of adverse events, dose modification, and withdrawal[*].**

| Adverse event | PegIFN alfa-2a (n = 75) | NA (n = 74) | P |
|---|---|---|---|
| Discontinuation | | | |
| For safety reasons[¶] | 10 (13.3) | 0 (0.0) | <0.001 |
| For other reasons | 2 (2.7) | 5 (6.8) | 0.276 |
| Dose modification | | | |
| Total | 5 (6.7) | 0 (0.0) | 0.058 |
| Adverse events | 0 (0.0) | 0 (0.0) | - |
| Laboratory abnormality | 5 (6.7) | 0 (0.0) | 0.058 |
| ≥1 adverse event | 64 (85.3) | 28 (37.8) | <0.001 |
| ≥1 serious adverse event | 3(4.0) | 2 (1.4) | 1.000 |
| Death | 0 (0.0) | 0 (0.0) | - |
| ALT flare due to CHB exacerbation[†] | 3 (18.7) | 1 (1.4) | 0.620 |
| Lung cancer | 0 (0.0) | 1 (1.4) | 0.497 |
| Maximum ALT level | | | |
| <ULN | 24 (32.0) | 50 (67.6) | <0.001 |
| 1–5×ULN | 56 (74.6) | 23 (31.1) | <0.001 |
| 5–10×ULN | 5 (6.7) | 0 (0.0) | 0.058 |
| >10×ULN | 1 (1.3) | 1 (1.4) | 1.000 |

[*]Values indicate non-cumulative numbers and percentages.

[¶] Decreased visual acuity (n = 1), pneumonia (n = 1), depression (n = 2), insomnia (n = 2), shortness of breath (n = 1), general ache (n = 1), weight loss (n = 1), itching sensation (n = 1)

[†] Defined as ALT>5×ULN.

CHB, chronic hepatitis B; ALT, alanine transaminase; ULN, upper limit of normal; PegIFNα-2a, peginterferon α-2a; NA, nucleos(t)ide analogues

**Table 5. Incidence of adverse events**[*].

| | PegIFN alfa-2a (n = 75) | NA (n = 74) |
|---|---|---|
| Adverse event leading to discontinuation | | |
| Weight loss | 1 (1.3) | 0 |
| Insomnia | 2 (2.6) | 0 |
| General ache | 1 (1.3) | 0 |
| Depression | 2 (2.6) | 0 |
| Shortness of breath | 1 (1.3) | 0 |
| Decreased visual acuity | 1 (1.3) | 0 |
| Pneumonia | 1 (1.3) | 0 |
| Itching sensation | 1 (1.3) | 0 |
| Any adverse events | | |
| ALT elevation | 62 (82.6) | 24 (32) |
| Flu-like symptom | 23 () | 0 |
| Neutropenia | 9 (12) | 0 |
| Alopecia | 5 (6.7) | 0 |
| Headache | 2 (2.6) | 0 |
| Fatigue | 4 (5.3) | 0 |
| Dyspepsia | 2 (5.3) | 0 |
| Itching sensation | 3 (5.3) | 0 |
| Depression | 1 (1.3) | 0 |
| Acute gastroenteritis | 0 | 1 (1.3) |
| Hypertension | 3 (2.6) | 1 (1.3) |
| Diabetes | 0 | 1 (1.3) |
| Hypothyroidism | 1 (1.3) | 0 |
| Ecchymosis | 1 (1.3) | 0 |
| Chest discomfort | 1 (1.3) | 0 |
| Thrombocytopenia | 2 (2.6) | 0 |
| Gingiva pain | 1 (1.3) | 0 |
| Pulmonary tuberculosis | 0 | 1 (1.3) |
| Upper respiratory infection | 2 (2.6) | 0 |
| Eczema | 1 (1.3) | 0 |

[*]Values are given as noncumulative number and percentage.

PegIFNα-2a, peginterferon α-2a; NA, nucleos(t)ide analogues

depression (n = 2), decreased visual acuity (n = 1), weight loss (n = 1), or itching sensation (n = 1). Six patients in the PegIFN alfa-2a group experienced ALT flares (>5×ULN), three within 12 weeks after beginning PegIFN alfa-2a treatment, and spontaneous resolution without dose reduction or discontinuation occurred in all 3 patients. The ALT flares in the other 3 patients, including 1 with a very high ALT level (>10×ULN), occurred after HBV DNA elevation; resolution occurred in all 3 patients after the addition of NA with concomitant PegIFN alfa-2a. One patient in the NA group who received 0.5 mg/day entecavir experienced an ALT flare that occurred after virologic breakthrough at week 12. This ALT flare resolved after addition of tenofovir (300 mg/day).

The rates of neutropenia and thrombocytopenia during antiviral treatment were higher in the PegIFN alfa-2a group (Table 5). Most of these incidents in the PegIFN alfa-2a group resolved after dose reduction, except for one patient who had concomitant pneumonia. This patient, who developed pneumonia with grade-3 neutropenia (520 cells/mm$^3$) after week 24,

discontinued PegIFN alfa-2a for 1 week, and resumed treatment after recovery from the pneumonia. This patient completed 48 weeks of PegIFN alfa-2a treatment without further adverse events.

## Discussion

This prospective, randomized, controlled study enrolled patients who were HBeAg-positive and did not experience HBeAg loss or seroconversion despite prior long-term NA treatment. We compared the on-treatment efficacy and safety of patients who switched to PegIFN alfa-2a with patients who continued NA for 48 weeks. Although several previous studies evaluated the effects of the combined use of NA and PegIFN alfa-2a, the optimal combination regimen remains unknown. Early trials indicated that *de novo* concurrent combinations did provide clinical benefit [12, 17], whereas addition of or switching to PegIFN alfa-2a therapy provided greater efficacy than NA alone in patients who had suppressed HBV DNA following NA treatment [18, 20]. However, these studies were performed in highly selected patients and reported efficacy in populations that had specific characteristics. The present study investigated the effect of switching to PegIFN alfa-2a by patients who received various NA maintenance therapies, but remained HBeAg-positive. Therefore, the results of our study are likely more relevant for CHB patients in actual clinical settings.

The results of this study showed that, compared with continuing NA for 48 weeks, switching to PegIFN alfa-2a for 48 weeks significantly reduced the HBsAg titer and increased HBeAg seroconversion in patients who achieved virological suppression following oral NA treatment. We used change in HBsAg concentration from baseline as the primary endpoint, because HBsAg concentration correlates with the level of intrahepatic covalently closed circular HBV DNA in HBeAg-positive patients [21]. The aim of this study was to evaluate the extent of HBsAg decline due to PegIFN alfa-2a therapy for a definite duration in patients who had CHB, were HBeAg-positive, and previously received long-term NA therapy. A significant decline of HBsAg can induce a durable viral response and ultimately a functional cure. We found that the maximal mean decline in HBsAg level from baseline to week 48 was $0.50 \pm 0.88$ $\log_{10}$ IU/mL in the PegIFN alfa-2a group, but only $0.08 \pm 0.46$ $\log_{10}$ IU/mL in NA group, a statistically significant difference. Another study reported that the mean decline in HBsAg level during 48 weeks of treatment with PegIFN alfa-2a plus entecavir was about $1.0$ $\log_{10}$ IU/mL, and that this decline persisted during follow-up when patients received entecavir alone [20]. However, most Korean CHB patients are infected with HBV genotype C, which has a poorer response to PegIFN alfa-2a than other genotypes [22, 23]. Furthermore, all of our enrolled patients were HBeAg-positive, despite long term NA treatment (mean duration > 5 years), making them a difficult-to-treat population. These factors may explain the smaller decline of HBsAg level in our patients.

At week 48, the rate of HBeAg seroconversion in our PegIFN alfa-2a group was 20%, but it was only 6.8% in our NA group (p = 0.018). A previous study that also examined the effect of switching therapy showed that the HBeAg seroconversion rate was 14.9 to 21% in a PegIFN alfa-2a group [18, 20]. Another study of treatment-naive HBeAg-positive CHB patients reported the HBeAg seroconversion rate was 29 to 32% after 48 weeks of PegIFN alfa-2a treatment and 12 to 22% after 1 year of various oral NA therapies [24]. Our study population consisted of difficult-to-treat patients (low HBeAg seroconversion rate following NA), and the rate of HBeAg seroconversion in our patients was comparable to those of previous studies. However, head-to-head comparisons are not possible due to differences in study design and population characteristics.

There were similar rates of adverse events in our study and previous studies. Although most of the patients in our PegIFN alfa-2a group finished the 48-week course of treatment without

severe adverse events, there were 10 severe adverse events, and 5 patients required permanent discontinuation due to safety issues. In addition, ALT elevation was more common in our PegIFN alfa-2a group (62/75 *vs*. 24/74). Among the 62 events of ALT elevation in our PegIFN alfa-2a group, 47 were associated with immunologic reactions to PegIFN alfa-2a. Previous research reported these ALT flares occurred in patients who responded to PegIFN alfa-2a, and were associated with immunologic clearance of HBV [25]. In the present study, ALT elevation in the PegIFN alfa-2a group was associated with a decline of HBsAg at week 48 (S4 Table), but not with HBeAg seroconversion at that time. The incidence of ALT elevation associated with breakthrough in our PegIFN alfa-2a group was significantly higher than reported in previous studies [18]. In addition, the rate of viral breakthrough was also slightly higher in our study, perhaps because all of our patients were HBeAg-positive at the time of enrollment and NA treatment was stopped after 3 months of PegIFN alfa-2a treatment. However, NA add-on treatment led to good control of viral breakthrough, and PegIFN alfa-2a was continued until week 48 without interruption.

This study had several limitations. HBV genotype could not be determined because the HBV DNA levels at baseline were too low. Thus, the effect of genotype on response to PegIFN alfa-2a could not be assessed. However, 99~100% of Korean patients infected with HBV are infected with genotype C [22], which is associated with more aggressive liver disease and poorer response to antiviral therapy than genotype B [23, 26, 27]. This might explain the smaller decline of HBsAg titer in our patients. Second, adding on PegIFN alfa-2a (rather than switching) could lead to higher response rates in similar populations without viral break-through, suggesting that inclusion of a PegIFN alfa-2a add-on group may have provided better information. Because this study was designed to investigate whether PegIFN alfa-2a alone could induce durable responses in patients with well-suppressed HBV DNA following long term NA treatment, we analyzed PegIFN alfa-2a switching rather than add-on. Third, the time point for rescue NA treatment in both groups could not be predefined because the need for adding NA as a rescue medication was not predictable. Fewer than half of our patients had restarted NA treatment by week 48. However, the impact on mean HBsAg level would be minor, because most of them restarted at week 48. Furthermore, adding NA as recue therapy did not affect mean HBsAg level, as shown in the S5 Table. Fourth, we collected 2 years of follow-up data after the 48 weeks of PegIFN alfa-2a switching therapy, as shown in the protocol. However, not all of the follow-up data were available at the time of writing. As a result, the secondary endpoint in this study was not exactly the same as described in the protocol. We will present these data soon. Fifth, because the primary endpoint was a continuous variable and not a dichotomous variable, it was difficult to account for loss-to-follow-up in the primary efficacy analysis. Thus, we used the LOCF approach for missing values in the continuous variables included in the efficacy evaluation. In addition, we performed sensitivity analysis to assess the robustness of the findings from the primary analysis. Finally, patients in this study had been treated with various types of oral NA and more than 10% of patients were treated with lamivudine-based regimens that are no longer used as first line NA. However, the use of previous types of oral NA did not affect the rate of viral breakthrough or HBeAg seroconversion (S7 Table) or the level of HBsAg decline (S4 Table).

In conclusion, this study of CHB patients with suppressed HBV DNA but with HBeAg positivity following NA treatment showed that switching to 48 weeks of PegIFN alfa-2a treatment, rather than continuing NA therapy, significantly reduced the HBsAg titer and increased rate of HBeAg seroconversion. However, HBV DNA was frequently elevated in patients who received PegIFN alfa-2a, thus warranting use of NA as an add-on treatment.

## Supporting information

**S1 Checklist. CONSORT 2010 checklist of information to include when reporting a randomised trial**[*].
(DOC)

**S1 Table. Outcome variables at each assessment time in the two groups.**
(DOCX)

**S2 Table. Outcomes at each assessment time in the two groups based on ITT with LOCF, ITT without OCF, and complete case analysis.**
(DOCX)

**S3 Table. Changes in serum HBV DNA levels in the two groups.**
(DOCX)

**S4 Table. Predictors of a decline in HBsAg by 0.5 $\log_{10}$ IU/mL or more at week-48 in all 149 patients.**
(DOCX)

**S5 Table. HBsAg levels in the 75 patients who received PegIFN alfa-2a alone or PegIFN alfa-2a with NA retreatment.**
(DOCX)

**S6 Table. Achievement of two different combined end-points by patients in the two groups.**
(DOCX)

**S7 Table. Prognostic factors associated with HBeAg seroconversion at week-48 in all 149 patients.**
(DOCX)

**S8 Table. Achievement of two different end-points by patients in the two groups.**
(DOCX)

**S1 Fig.** Changes of HBsAg level (A), HBsAg reduction (B), and HBeAg seroconversion (C) in patients with ALT elevation due to viral breakthrough or due to use of PegIFN alfa-2a. HBsAg, Hepatis B surface Antigen; ALT, alanine aminotransferase; HBeAg, hepatitis B e Antigen; PegIFNα-2a, peginterferon α-2a.
(TIF)

**S2 Fig.** Change in ALT level (A) during treatment for the PegIFN and NA groups and changes in HBsAg level (B) and ALT level (C) according to HBeAg seroconversion at week 48. HBsAg, Hepatis B surface Antigen; ALT, alanine aminotransferase; NA, nucleos(t)ide analogues; PegIFNα-2a, peginterferon α-2a. [*] $p < .05$, [***] $p < .001$.
(TIF)

**S1 Dataset.**
(XLSX)

**S1 File.**
(PDF)

**S2 File.**
(PDF)

## Author Contributions

**Conceptualization:** Jeong Heo, Won Young Tak.

**Data curation:** Hyun Young Woo, Jeong Heo, Won Young Tak, Heon Ju Lee, Woo Jin Chung, Jung Gil Park, Soo Young Park, Young Joo Park, Yu Rim Lee, Jae Seok Hwang, Young Oh Kweon.

**Formal analysis:** Hyun Young Woo, Jeong Heo, Won Young Tak.

**Funding acquisition:** Jeong Heo.

**Investigation:** Hyun Young Woo, Jeong Heo, Won Young Tak, Soo Young Park, Young Joo Park, Yu Rim Lee.

**Methodology:** Hyun Young Woo, Won Young Tak, Soo Young Park.

**Project administration:** Hyun Young Woo.

**Resources:** Hyun Young Woo, Jeong Heo, Won Young Tak, Heon Ju Lee, Woo Jin Chung, Jung Gil Park, Jae Seok Hwang, Young Oh Kweon.

**Software:** Hyun Young Woo, Won Young Tak.

**Supervision:** Jeong Heo, Won Young Tak, Heon Ju Lee, Soo Young Park, Young Oh Kweon.

**Validation:** Jeong Heo, Won Young Tak, Soo Young Park.

**Visualization:** Jeong Heo, Won Young Tak, Soo Young Park.

**Writing – original draft:** Hyun Young Woo.

**Writing – review & editing:** Hyun Young Woo, Jeong Heo, Won Young Tak, Soo Young Park.

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
