## [Decision Letter · Decision Letter 0]

21 Feb 2022

PONE-D-22-01751Switching to PegIFN alfa-2a in HBeAg-positive chronic hepatitis B patients undergoing nucleos(t)ide maintenance therapy: A randomized trialPLOS ONE

Dear Dr. Heo,

Thank you for submitting your manuscript to PLOS ONE. After careful consideration, we feel that it has merit but does not fully meet PLOS ONE’s publication criteria as it currently stands. Therefore, we invite you to submit a revised version of the manuscript that addresses the points raised during the review process.

We look forward to receiving your revised manuscript.

Kind regards,

Jee-Fu Huang, M.D., Ph.D.

Academic Editor

PLOS ONE

Journal Requirements:

2. Thank you for submitting your clinical trial to PLOS ONE and for providing the name of the registry and the registration number. The information in the registry entry suggests that your trial was registered after patient recruitment began. PLOS ONE strongly encourages authors to register all trials before recruiting the first participant in a study.

1) your reasons for your delay in registering this study (after enrolment of participants started);

2) confirmation that all related trials are registered by stating: “The authors confirm that all ongoing and related trials for this drug/intervention are registered”.

I have read the journal's policy and the authors of this manuscript have the following competing interests: [JH received a grant from Roche Korea, HYW, WYT, HJL, WJC, JGP, YJP, YRL, SYP, YOK and JSH report no conflicts of interest.]. 

Reviewers' comments:

Reviewer's Responses to Questions

5. Review Comments to the Author

Reviewer #1: The present study aims to investigate change in HBsAg titer in chronic hepatitis B patients switching to PegIFN alfa-2a after long term NA therapy. This is a well design, randomized trial. However, this issue had been well studied before and the present study does not provide any novel finding. Comments to this study list below.

1. The inclusion criteria included undetectable HBV viral load, which was defined as HBV DNA ≤100 IU/mL, for at least 12 months. HBV DNA of 100 IU/mL as undetectable is relatively too high. A lower level of HBV DNA would be more indicated. Besides, the efficacy analysis of undetectable HBV viral load was defined as HBV DNA ≤20 IU/mL. (Line 204) Can authors explain why a different definition of undetectable HBV viral load? And I suggest a uniform definition for this.

2. As we know, treatment response may develop even after stopping PegIFN therapy. Did authors analyze post-IFN response, like change of HBsAg titer/HBsAg loss after stopping PegIFN?

3. A large proportion of patients were treated with lamivudine-based regimens which was no more used as first line NA. This would be one of the major limitations. And for those developed viral breakthrough, were they developed YMDD mutation?

4. Table 1 - Please check HBV DNA data and confirm data accuracy.

5. How many patients had HBsAg loss? And whether a difference of HBsAg loss between 2 groups?

6. Table 4 – at 48 weeks, only 6 (8.0%) of patients in PegIFN group had HBV DNA ≥20 and 2000 IU/mL, but 10 (13.3%) patients had HBV DNA ≥20000 IU/mL. Is the data correct? Authors should check the data again.

7. Table 6 - Table format needs modification for a better reading.

8. Table 7 – the number of ALT elevation in PegIFN group was 62 (82.6%), but the same ALT elevation showed 51 (68.0%) in table 6. Authors should explain this inconsistent data and confirm which one is correct.

Reviewer #2: Woo et al compared the HBsAg levels and decline and HBeAg seroconversion rates in patients with HBeAg-positive CHB who had been virally suppressed with nucleos(t)ide analogues (NAs) and switched to PegIFN alfa-2a or continued NA for another 48 weeks in a prospective, randomized multi-center study. They demonstrated that PegIFN alfa-2a treatment yielded a significantly greater decline in HBsAg levels and a significantly higher rate of HBeAg seroconversion both at 24, 36 and 48 weeks of treatment. Patients had relatively good tolerance to PegIFN alfa-2a treatment. This study provides valuable data on the extent of HBsAg decline and HBeAg seroconversion rate in patients with HBeAg-positive CHB who started NA therapy and already achieved HBeAg seroconversion with undetectable HBV DNA and switch to PegIFN alfa-2a with an aim to achieving HBsAg loss. There are several issues the authors need to address to improve the scientific merit of the manuscript.

1. Page 7, line 160: undetectable HBV DNA defined as HBV DNA ≤100 IU/mL versus Page 9, line 204: HBV DNA <20 IU/mL. Please clarify.

2. Page 7, line 168- Page 8, line 180 on “exclusion criteria”: some of the sentences look like being copied from the informed consent form. Please correct.

3. Table 2: Are HBsAg levels and reductions at 12, 24, 36 and 48 weeks after switching to PegIFN alfa-2a derived from all 75 patients? Because a significant proportion of patients received NA retreatment (1.3%, 9.3%, 4.0% and 32% at 12, 24, 36 and 48 weeks, respectively, Table 5), it is worthwhile to further address whether there is a difference in HBsAg levels at various time points between patients who received PegIFN alfa-2a alone and those who received NA retreatment. This information may unveil the impact of NA retreatment on HBsAg decline in case of viral breakthrough. Please also discuss this finding as appropriate.

4. Table 4: At 48 weeks of PegIFN alfa-2a treatment, there were 6 (8.0%) patients with HBV DNA ≥20 IU/mL or ≥2000 IU/mL. Why were there 10 (13.3%) patients with HBV DNA ≥20000 IU/mL? Please clarify.

5. Table 5: 24 (32%) patients received NA retreatment at 48 weeks of PegIFN alfa-2a treatment. Why? What are the indications?

6. Page 20, line 364- Page 21, line 373: 15 incidents of ALT elevation occurred following viral breakthrough whereas 47 incidents of ALT elevation occurred during the first 3 months of PegIFN alfa-2a treatment while the patients were still receiving NA therapy. Please compare HBsAg levels and reductions and HBeAg seroconversion rates across treatment course between these two subgroups of patients and present the data at least in a supplementary figure, similar to Figure 2.

7. Table 6: 23 (31.1%) patients had maximal ALT levels 1-5x ULN in the NA group, which was not attributed to viral breakthrough. What are the causes of these ALT elevations? Please clarify and discuss.

8. Some typos and grammar mistakes. This manuscript needs English editing.

Reviewer #3: This is an important RCT that provides the information that switching to PegIFN from NA is beneficial for HBeAg-positive patients who received NA treatment for 72 weeks at least. The switch group has a higher HBeAg seroconversion rate and a more prominent HBsAg reduction.

I have a few comments as follows

1. The English writing could be improved. For example, the authors do not need to emphasize “significant” when doing some comparison. It is clear to achieve a statistic difference when the p-value is present. In addition, the following sentence present in the exclusion criteria is inappropriate “if you have been given an immunomodulatory/immunosuppressant within 6 months prior to registration “

2. It is better to report the data of combined endpoint, for example HBeAg loss + viral load < 2000 IU/mL

3. Previous data has shown that HBsAg level <100 IU/mL is associated with higher HBsAg loss in patients with spontaneous HBeAg seroconversion.1 Please define HBeAg seroconversion + HBsAg <100 IU/mL as another combined endpoint and present the data about the patient number achieving this endpoint in each group.

Reference

1. Tseng TC, Liu CJ, Su TH, et al. Serum hepatitis B surface antigen levels predict surface antigen loss in hepatitis B e antigen seroconverters. Gastroenterology 2011;141:517-525 e512.

Reviewer #4: 

# Major Issues

## Introduction

The concept of *durable* viral response appears twice in the first two sentences of the abstract and elsewhere in the text yet is *not* defined. This must be corrected. What do the authors mean by a "durable" viral response? Is there a standard or accepted definition for this phenomenon?

## Methods

The description of the inclusion and exclusion criteria need to be revised. There are many inconsistencies with tenses (some verbs use the present tense while others use the past tense), person (some statements use the second person and others use the third), voice (some statements use the passive voice while others use the active voice), and clause structure (some a clauses while others are complete sentences). The authors need to be consistent and apply grammatical rules appropriately.

*IMPORTANT* Lines 181-196 describe the use of medication to reduce ALT flares and in the event of viral breakthrough. However, this only applies to those receiving PegIFN. This is in direct contradiction to the design stated in the objectives (Lines 130-132) where PegIFN was compared to NA for 48 weeks. Formally, the authors are NOT testing PegIFN versus NA, but PegIFN plus NA for 12 weeks then PegIFN with NA as rescue medication for 36 weeks versus NA alone for 48 weeks. You can see how the second description is quite unlike the first. The authors need to be transparent about the two arms. This has to be clear from the title and abstract.

*IMPORTANT* Why is there no commensurate rescue management for viral breakthrough in those receiving NA alone?

*IMPORTANT* Why are the definitions of viral breakthrough dependent on the medication group? What justification is there for this?

*IMPORTANT* In Line 195, the authors state two terms -- viral breakthrough and viral rebound. Only the former is defined. The latter needs to be defined, too.

*IMPORTANT* There is a major discrepancy between the primary outcome as stated in the protocol compared to the outcome defined in the manuscript. In the protocol, the primary endpoint is defined as "Primary endpoint: Changes in HBsAg quantity (log10 HBsAg) during administration of antiviral

agents in each group".

*IMPORTANT* There appears to be discrepancies between the secondary endpoints as stated in the protocol and the ones reported here:

1. The protocol states that secondary endpoint 1 is changes from baseline in serum HBV DNA levels and HBV DNA non-detection rates and HBV DNA <20 IU/mL during administration of antiviral agents in each group and follow-up, comparison of the ratio of HBV DNA <2,000 IU/mL during follow-up and administration of antiviral agents of each group and comparison of the ratio of HBV DNA <20,000 IU/mL during the administration of antiviral agents of each group and follow-up. In contrast, the present manuscript (Lines 204-205) only report the proportions of MBV DNA <20, <2,000 and >20,000 IU/mL.

2. The protocol states that secondary endpoint 2 is HBeAg seroconversion rate and loss rate, in contrast to the manuscript, which only reports HBeAg seroconversion.

3. The manuscript identifies secondary endpoint 3 as HBsAg loss and HBsAg seroconversion at the end of treatment. This appears nowhere in the protocol. The planned secondary endpoint is actually HBsAg loss and HBsAg seroconversion 1 and 2 years after administration/termination of antiviral agents in each group.

4. Secondary endpoint 4 in the protocol is not included in this manuscript.

5. This manuscript includes an examination of predictors of seroconversion, which does not appear in the protocol.

The authors MUST adhere to the protocol. All endpoints must be presented as planned. All analyses that do not appear in the protocol are necessary post-hoc and must be deleted or identified as such. All deviations from the protocol MUST be identified and justified. This is non-negotiable.

*IMPORTANT* The description of the safety endpoints is quite inadequate. Significantly, the authors claim that "serious" adverse events were a criterion for early dropout of participants, but the term "serious" is not defined (although it is in the protocol).

*IMPORTANT* The authors use the "last observation carried forward" (LOCF) method to impute missing values. The LOCF method has been shown many. many times to be of dubious statistical validity. The use of this method has been discouraged due to the biases that may arise. The authors have adopted the method outside of the parameters of the protocol. Thus, the LOCF method should NOT be used here. The authors must perform the analysis without the LOCF method. If they wish to apply the LOCF method, then they can do so under sensitivity analysis. This is non-negotiable.

*IMPORTANT* Both the linear mixed models and the general estimating equation models must be described in sufficient detail to enable replication. The authors need to enhance their description of these methods as important information is missing.

*IMPORTANT* The ethics approval provided by the authors allowed only for the recruitment of 144 patients. In contrast, the authors randomised 149 patients, or five more than they were allowed. The authors need to provide evidence that this modification was submitted to the ethics committee and permission was obtained.

## Results

Lines 281-282: "All patients were... and at randomisation." This sentence is redundant because this status is required for study entry. This can be deleted.

The section headed "Predictors of HBeAg seroconversion at week 48" should be deleted. This is NOT an objective stated in the protocol and is entirely post-hoc.

## Table 1

Line 289 should be deleted and the information should be included in the stub.

The column of p-values should be omitted. It is highly inappropriate to apply formal statistical tests for baseline characteristics in a randomised controlled trial as this is meaningless.

## Table 2

The information here is better presented as a graph.

Line 305 should be deleted and the information should be included in the stub.

Line 313 should be deleted.

*IMPORTANT* The results of the linear mixed model MUST adjust for the baseline level of the outcome. It is not clear if this is the case.

## Table 4

The subheadings are wrong. They must be changed.

## Table 5

Standard deviations should be provided, not standard errors. Alternatively, 95% confidence intervals can be provided in place of standard errors.

## Table 6

The footnote explaining that the chi-square test could not be used is wrong. This is because the data presented are wrong. The authors need to provide the number of patients in each group who experienced at least one adverse event. This number cannot exceed the total number of participants. This must be corrected.

# Minor Issues

The authors need to define all abbreviations and acronyms on first use. HBsAg is defined, but HBeAg is not, for example. They need to check the manuscript for similar occurrences.

The demonstrative pronoun in Line 130 is vague. I read it to mean reference 18. Please improve the sentence to reduce confusion.

The compound adjective in Line 132 is missing a hyphen, as are those in Lines 158-159 and elsewhere. Please review and modify.

What is the limit of detection of HBeAg? That is to say, at what titer of HBeAg is a patient considered HBeAg-positive?

Line 238: replace "numeric" with "continuous"

Line 247: replace "observed" with "observation"

# Recommendation

The clinical trial described in the manuscript contains many unexplained and important deviations from the protocol. I am unable to support the approval of this manuscript for publication in the journal until these issues are corrected.

Reviewer #5: This study by Heo J et al provides important information for how the peg-IFN may benefit on the the HBeAg positive CHB patients who remained HBeAg positive after 5-6 years of NA treatment. This is a well conduct study.

There are few questions remained to be answered before acceptance:

1. When the ALT elevation occurred in the peg-IFN arm? The on-treatment ALT flare is worth to be noted especially when the higher HBeAg seroconversion in the peg-IFN group. Have these patients checked with the autoimmune hepatitis markers to exclude the possibility of IFN related autoimmune issue? It would be great if the ALT kinetics be provided in addition to HBsAg kinetics as well.

2. What's the reason for ALT elevation in the NA treated arm? It is quite surprising that the proportion for ALT > 1X ULN in the NA arn is as high as 32.4%!

3. How many proportion of these two arms patients had reach HBsAg quantification level < 100 by the end of the study? How many proportion reach HBsAg rapid decline by the end of the study, which defined as HBsAg reduction > 0.5 log10IU/mL per year? Since these two endpoints may be the sentinal predictor for subsequent HBsAg loss, it is important to know whether the peg-IFN arm have the higher chance for functional cure.

4. What's the outcome of the dropped out patients? It is also interesting to know what happened when these treated 5-6 years patients be discontinued from antiviral therapy.

---

## [Author Response · Author response to Decision Letter 0]

21 Apr 2022

Journal Requirements:

As indicated, our manuscript meets PLOS ONE's style requirements. 

2. Thank you for submitting your clinical trial to PLOS ONE and for providing the name of the registry and the registration number. The information in the registry entry suggests that your trial was registered after patient recruitment began. PLOS ONE strongly encourages authors to register all trials before recruiting the first participant in a study.

1) your reasons for your delay in registering this study (after enrolment of participants started);

This study was planned at January 2012 and registered to clinicaltrial.gov at January 2013 after IRB approval. Patient enrollment started from August 2013. 

2) confirmation that all related trials are registered by stating: “The authors confirm that all ongoing and related trials for this drug/intervention are registered”.

 We confirmed this statement.

I have read the journal's policy and the authors of this manuscript have the following competing interests: [JH received a grant from Roche Korea, HYW, WYT, HJL, WJC, JGP, YJP, YRL, SYP, YOK and JSH report no conflicts of interest.]. 

We have included an updated competing interests statement in the cover letter.

We understand and will update the data availability statement.

As indicated, we provided these data as supplementary information.

Reviewers' comments:

Reviewer's Responses to Questions

5. Review Comments to the Author

Reviewer #1: The present study aims to investigate change in HBsAg titer in chronic hepatitis B patients switching to PegIFN alfa-2a after long term NA therapy. This is a well design, randomized trial. However, this issue had been well studied before and the present study does not provide any novel finding. Comments to this study list below.

1. The inclusion criteria included undetectable HBV viral load, which was defined as HBV DNA ≤100 IU/mL, for at least 12 months. HBV DNA of 100 IU/mL as undetectable is relatively too high. A lower level of HBV DNA would be more indicated. Besides, the efficacy analysis of undetectable HBV viral load was defined as HBV DNA ≤20 IU/mL. (Line 204) Can authors explain why a different definition of undetectable HBV viral load? And I suggest a uniform definition for this.

Thank you for the comment. At the beginning of this trial in 2012, the detection limit of HBV DNA was 100 IU/mL, but the detection limit at present is 10 or 20 IU/mL. We therefore revised the Methods to say “... had an HBV DNA level of 100 IU/mL or less for at least 12 months (the detection limit at study onset in 2012).” We considered the limit of detection to be 20 IU/mL.

2. As we know, treatment response may develop even after stopping PegIFN therapy. Did authors analyze post-IFN response, like change of HBsAg titer/HBsAg loss after stopping PegIFN?

This is a good point. We collected patient data for 2 years after recording the IFN response, but these results are not yet available. We plan to present these data soon.

3. A large proportion of patients were treated with lamivudine-based regimens which was no more used as first line NA. This would be one of the major limitations. And for those developed viral breakthrough, were they developed YMDD mutation?

Patients randomized to the PegIFN alfa-2a group continued taking oral NAs for the first 12 weeks of PegIFN alfa-2a treatment. After that, patients in PegIFN alfa-2a stopped taking oral NAs. The rate of viral breakthrough in the PegIFN alfa-2a group was similar in those who initially used different oral NAs (entecavir: 32.1%, 9/28; tenofovir: 50%, 2/4; lamivudine: 50%, 1/2; lamivudine+adefovir: 50%, 6/12; entecavir+adefovir: 35.7%, 5/14; entecavir+tenofovir: 63.7%, 7/11; lamivudine+tenofovir: 100%, 4/4; P = 0.176) One patient who received lamivudine developed viral breakthrough at week-36, had a YMDD mutation, and received concomitant tenofovir. In patients randomized to the oral NA group, viral breakthrough did not occur in patients using lamivudine alone, and only occurred in 1 patient who received entecavir (0.5 mg).

4. Table 1 - Please check HBV DNA data and confirm data accuracy.

The data are accurate and expressed as log10(IU/mL) of HBV DNA.

5. How many patients had HBsAg loss? And whether a difference of HBsAg loss between 2 groups? 

One patient in the PegIFN alfa-2a group had HBsAg loss at week-48. No patients in NA group had HBsAg loss. We added this information to the Results.

6. Table 4 – at 48 weeks, only 6 (8.0%) of patients in PegIFN group had HBV DNA ≥20 and 2000 IU/mL, but 10 (13.3%) patients had HBV DNA ≥20000 IU/mL. Is the data correct? Authors should check the data again.

This is a good point. The data are correct. Six patients who had HBV DNA above 2000 IU/mL but less than 20,000 IU/mL at week-24 and week-36 developed HBV DNA above 20,000 IU/mL at week-48. Rescue oral antiviral therapy for breakthrough was delayed in these patients. 

7. Table 6 - Table format needs modification for a better reading.

We revised the Table (now Table 4) as indicated. 

8. Table 7 – the number of ALT elevation in PegIFN group was 62 (82.6%), but the same ALT elevation showed 51 (68.0%) in table 6. Authors should explain this inconsistent data and confirm which one is correct.

This is good point. The number 51 was wrong in Table 6, so we made the correction. The number of adverse events was also presented in Table 7 (now Table 5), so we revised Table 6 (now Table 4) by deleting the last column of this table. 

Reviewer #2: Woo et al compared the HBsAg levels and decline and HBeAg seroconversion rates in patients with HBeAg-positive CHB who had been virally suppressed with nucleos(t)ide analogues (NAs) and switched to PegIFN alfa-2a or continued NA for another 48 weeks in a prospective, randomized multi-center study. They demonstrated that PegIFN alfa-2a treatment yielded a significantly greater decline in HBsAg levels and a significantly higher rate of HBeAg seroconversion both at 24, 36 and 48 weeks of treatment. Patients had relatively good tolerance to PegIFN alfa-2a treatment. This study provides valuable data on the extent of HBsAg decline and HBeAg seroconversion rate in patients with HBeAg-positive CHB who started NA therapy and already achieved HBeAg seroconversion with undetectable HBV DNA and switch to PegIFN alfa-2a with an aim to achieving HBsAg loss. There are several issues the authors need to address to improve the scientific merit of the manuscript.

1. Page 7, line 160: undetectable HBV DNA defined as HBV DNA ≤100 IU/mL versus Page 9, line 204: HBV DNA <20 IU/mL. Please clarify.

Thank you for the excellent comment, which was also noted by Review #1. At the beginning of this trial in 2012, the detection limit of HBV DNA was 100 IU/mL, but the detection limit at present is 10 or 20 IU/mL. We therefore revised the Methods to say “... had an HBV DNA level of 100 IU/mL or less for at least 12 months (the detection limit at study onset in 2012).” We considered the limit of detection to be 20 IU/mL..

2. Page 7, line 168- Page 8, line 180 on “exclusion criteria”: some of the sentences look like being copied from the informed consent form. Please correct.

As indicated, we modified the exclusion criteria accordingly.

3. Table 2: Are HBsAg levels and reductions at 12, 24, 36 and 48 weeks after switching to PegIFN alfa-2a derived from all 75 patients? Because a significant proportion of patients received NA retreatment (1.3%, 9.3%, 4.0% and 32% at 12, 24, 36 and 48 weeks, respectively, Table 5), it is worthwhile to further address whether there is a difference in HBsAg levels at various time points between patients who received PegIFN alfa-2a alone and those who received NA retreatment. This information may unveil the impact of NA retreatment on HBsAg decline in case of viral breakthrough. Please also discuss this finding as appropriate.

This is a good point. As indicated, HBsAg level and reduction at 12, 24, 36, and 48 weeks were derived from all 75 patients. We presented data about HBsAg level at various times for patients who received PegIFN alfa-2a alone and those who received NA retreatment in a S5 Table.

4. Table 4: At 48 weeks of PegIFN alfa-2a treatment, there were 6 (8.0%) patients with HBV DNA ≥20 IU/mL or ≥2000 IU/mL. Why were there 10 (13.3%) patients with HBV DNA ≥20000 IU/mL? Please clarify.

This is a good point, which was also noted by Reviewer #1. The data are correct. Six patients who had HBV DNA above 2000 IU/mL but less than 20,000 IU/mL at week-24 and week-36 developed HBV DNA above 20,000 IU/mL at week-48. Rescue oral antiviral therapy for breakthrough was delayed in these patients. 

5. Table 5: 24 (32%) patients received NA retreatment at 48 weeks of PegIFN alfa-2a treatment. Why? What are the indications?

The indications for restarting NA treatment for patients in the PegIFN alfa-2a group was provided in the Methods: “Patients in the PegIFN alfa-2a group who experienced HBV DNA elevation (viral breakthrough), with or without ALT flare, were allowed to restart concomitant oral NA at the discretion of the researchers. Viral breakthrough in the PegIFN alfa-2a group was defined as an increase in HBV DNA to 2000 IU/mL or more during treatment.”

6. Page 20, line 364- Page 21, line 373: 15 incidents of ALT elevation occurred following viral breakthrough whereas 47 incidents of ALT elevation occurred during the first 3 months of PegIFN alfa-2a treatment while the patients were still receiving NA therapy. Please compare HBsAg levels and reductions and HBeAg seroconversion rates across treatment course between these two subgroups of patients and present the data at least in a supplementary figure, similar to Figure 2.

There was some overlap of patients (n=12) among those with ALT elevation due to viral breakthrough and ALT elevation during the first 3 months of PegIFN alfa-2a treatment. Therefore, we cannot perform a statistical comparison of these groups. Instead, we presented the HBsAg level, HBsAg reduction, and HBeAg seroconversion during treatment in these two subgroups of patients as a S1 Fig. 

7. Table 6: 23 (31.1%) patients had maximal ALT levels 1-5x ULN in the NA group, which was not attributed to viral breakthrough. What are the causes of these ALT elevations? Please clarify and discuss.

Although the number of patients with ALT elevation in the NA group was not small, the ALT elevation was mild and not persistent. The causes of mild ALT elevation in the NA group are nonspecific, and may be due to use of a cold medication, excessive exercise, or excessive work. 

8. Some typos and grammar mistakes. This manuscript needs English editing.

 As indicated, English editing was performed.

Reviewer #3: This is an important RCT that provides the information that switching to PegIFN from NA is beneficial for HBeAg-positive patients who received NA treatment for 72 weeks at least. The switch group has a higher HBeAg seroconversion rate and a more prominent HBsAg reduction.

I have a few comments as follows

1. The English writing could be improved. For example, the authors do not need to emphasize “significant” when doing some comparison. It is clear to achieve a statistic difference when the p-value is present. In addition, the following sentence present in the exclusion criteria is inappropriate “if you have been given an immunomodulatory/immunosuppressant within 6 months prior to registration “

As indicated, English editing was performed.

2. It is better to report the data of combined endpoint, for example HBeAg loss + viral load < 2000 IU/mL

We presented these data in a S6 Table because this is not in the protocol.

3. Previous data has shown that HBsAg level <100 IU/mL is associated with higher HBsAg loss in patients with spontaneous HBeAg seroconversion.1 Please define HBeAg seroconversion + HBsAg <100 IU/mL as another combined endpoint and present the data about the patient number achieving this endpoint in each group.

We presented these data in a S6 Table because this is not in the protocol. However, the number of patients with HBeAg seroconversion plus HBsAg below 100 IU/mL was very small, and these patients did not show HBsAg loss until week-48. 

Reference

1. Tseng TC, Liu CJ, Su TH, et al. Serum hepatitis B surface antigen levels predict surface antigen loss in hepatitis B e antigen seroconverters. Gastroenterology 2011;141:517-525 e512.

Reviewer #4: 

# Major Issues

## Introduction

The concept of *durable* viral response appears twice in the first two sentences of the abstract and elsewhere in the text yet is *not* defined. This must be corrected. What do the authors mean by a "durable" viral response? Is there a standard or accepted definition for this phenomenon?

Durable viral response means sustained off-treatment response following NA cessation, such as HBs seroclearance and HBeAg seroconversion with undetectable HBV DNA. As indicated, we defined durable viral response in the Introduction as follows: “However, even after long-term NA treatment of these patients, many of them do not achieve a durable viral response (i.e., a sustained response following NA cessation), such as seroclearance of HB surface antigen (HBsAg), seroconversion of HB e-antigen (HBeAg), and undetectable HBV DNA.”

Reference: Liaw YF, Kao JH, Piratvisuth T, Chan HL, Chien RN, Liu CJ, et al. Asian-Pacific consensus statement on the management of chronic hepatitis B: a 2012 update. Hepatol Int. 2012;6:531–561.

## Methods

The description of the inclusion and exclusion criteria need to be revised. There are many inconsistencies with tenses (some verbs use the present tense while others use the past tense), person (some statements use the second person and others use the third), voice (some statements use the passive voice while others use the active voice), and clause structure (some a clauses while others are complete sentences). The authors need to be consistent and apply grammatical rules appropriately.

As indicated, English editing was performed.

*IMPORTANT* Lines 181-196 describe the use of medication to reduce ALT flares and in the event of viral breakthrough. However, this only applies to those receiving PegIFN. This is in direct contradiction to the design stated in the objectives (Lines 130-132) where PegIFN was compared to NA for 48 weeks. Formally, the authors are NOT testing PegIFN versus NA, but PegIFN plus NA for 12 weeks then PegIFN with NA as rescue medication for 36 weeks versus NA alone for 48 weeks. You can see how the second description is quite unlike the first. The authors need to be transparent about the two arms. This has to be clear from the title and abstract.

This is a very good point. However, the main purpose of this study was to evaluate the effect of switching from NA to PegIFN alfa-2a for 48 weeks compared to continuing NA. Adding NA for first 12 weeks was performed to reduce the risk of ALT flare that may be caused by switching to PegIFN alfa-2a, as described in the protocol. In the previous study, NA was also added for first 8 weeks. 

The need for adding NA as a rescue medication is not predictable, and only 11 patients needed retreatment prior to week-48; the other 24 patients started NA retreatment at week-48 (after the end of PegIFN alfa-2a treatment). Therefore, we conclude that adding NA as recue therapy did not affect the results, as shown in the S5 Table. Although this is an important detail, adding this information to the Title or Abstract could create confusion. 

*IMPORTANT* Why is there no commensurate rescue management for viral breakthrough in those receiving NA alone?

In case of viral breakthrough in the NA group, recue management such as adding a new NA or switching to another NA, was performed according to contemporary HBV guidelines, and considering the mutation profile and patient compliance at the discretion of the researchers. We added this in the Methods.

*IMPORTANT* Why are the definitions of viral breakthrough dependent on the medication group? What justification is there for this?

This is because virologic response of PegIFN alfa-2a is different from virologic response of NA. For PegIFN alfa-2a, virologic response is defined as a decrease in serum HBV DNA to below 2000 IU/mL after 6 months and at the end of therapy. For NA, virologic response is defined as a decrease in serum HBV DNA to an undetectable level based on a real-time PCR assay. 

*IMPORTANT* In Line 195, the authors state two terms -- viral breakthrough and viral rebound. Only the former is defined. The latter needs to be defined, too.

This is a good point. Viral rebound does not apply in this study, so we removed this text.

*IMPORTANT* There is a major discrepancy between the primary outcome as stated in the protocol compared to the outcome defined in the manuscript. In the protocol, the primary endpoint is defined as "Primary endpoint: Changes in HBsAg quantity (log10 HBsAg) during administration of antiviral agents in each group".

We have described primary endpoint of this study as the change in log10 HBsAg titer during antiviral therapy. We modified the text to say: “The primary endpoint was the change in HBsAg quantity (log10 HBsAg) during drug administration in each group. ” 

*IMPORTANT* There appears to be discrepancies between the secondary endpoints as stated in the protocol and the ones reported here:

1. The protocol states that secondary endpoint 1 is changes from baseline in serum HBV DNA levels and HBV DNA non-detection rates and HBV DNA <20 IU/mL during administration of antiviral agents in each group and follow-up, comparison of the ratio of HBV DNA <2,000 IU/mL during follow-up and administration of antiviral agents of each group and comparison of the ratio of HBV DNA <20,000 IU/mL during the administration of antiviral agents of each group and follow-up. In contrast, the present manuscript (Lines 204-205) only report the proportions of MBV DNA <20, <2,000 and >20,000 IU/mL.

This is a very good point. As indicated, this secondary endpoint was not exactly same as described in the protocol because we do not yet have all the follow-up data. Therefore, we deleted the term “follow-up”. We presented the change of HBV DNA level from baseline during 48 weeks of administration of antiviral agents in each group. Therefore, we corrected this text as follows: “The secondary endpoints were (i) changes in serum HBV DNA level from baseline and the ratio of below 20 IU/mL (the detection limit), below 2000 IU/mL, and below 20,000 IU/mL during drug administration in each group and (ii) change of HBeAg seroconversion and loss during drug administration in each group. 

2. The protocol states that secondary endpoint 2 is HBeAg seroconversion rate and loss rate, in contrast to the manuscript, which only reports HBeAg seroconversion.

As indicated, above, we corrected this text.

3. The manuscript identifies secondary endpoint 3 as HBsAg loss and HBsAg seroconversion at the end of treatment. This appears nowhere in the protocol. The planned secondary endpoint is actually HBsAg loss and HBsAg seroconversion 1 and 2 years after administration/termination of antiviral agents in each group.

As described above, we do not yet have all the follow up data. Therefore, we deleted this secondary endpoint.

4. Secondary endpoint 4 in the protocol is not included in this manuscript.

As described above, we do not yet have all the follow up data, so we did not include this secondary endpoint.

5. This manuscript includes an examination of predictors of seroconversion, which does not appear in the protocol.

This was a post-hoc analysis. Although it was not included as an endpoint, it is essential to analyze predictors associated with endpoints. We moved this text to the supplementary section (S7 Table).

The authors MUST adhere to the protocol. All endpoints must be presented as planned. All analyses that do not appear in the protocol are necessary post-hoc and must be deleted or identified as such. All deviations from the protocol MUST be identified and justified. This is non-negotiable.

*IMPORTANT* The description of the safety endpoints is quite inadequate. Significantly, the authors claim that "serious" adverse events were a criterion for early dropout of participants, but the term "serious" is not defined (although it is in the protocol).

This is a very good point. As indicated, it is not serious adverse event but severe adverse event. We added following text in the Methods: “A severe adverse event was defined as an event that led to study drug discontinuation.” And we modified the Table 6 (now Table 4) according to the definition of serious adverse event in the protocol

*IMPORTANT* The authors use the "last observation carried forward" (LOCF) method to impute missing values. The LOCF method has been shown many. many times to be of dubious statistical validity. The use of this method has been discouraged due to the biases that may arise. The authors have adopted the method outside of the parameters of the protocol. Thus, the LOCF method should NOT be used here. The authors must perform the analysis without the LOCF method. If they wish to apply the LOCF method, then they can do so under sensitivity analysis. This is non-negotiable.

Because the primary endpoint was not a dichotomous variable, it is difficult to account for loss-to-follow up in the PegIFN group, such discontinuation of this drug due to adverse events. Thus, we used the LOCF approach. Instead, we have performed sensitivity analysis, as described in the Methods and presented in a supplementary table. 

*IMPORTANT* Both the linear mixed models and the general estimating equation models must be described in sufficient detail to enable replication. The authors need to enhance their description of these methods as important information is missing.

As indicated, we modified the Methods section accordingly. 

*IMPORTANT* The ethics approval provided by the authors allowed only for the recruitment of 144 patients. In contrast, the authors randomised 149 patients, or five more than they were allowed. The authors need to provide evidence that this modification was submitted to the ethics committee and permission was obtained.

We reported this to the IRB and received permission to enroll 5 more patients.

## Results

Lines 281-282: "All patients were... and at randomisation." This sentence is redundant because this status is required for study entry. This can be deleted.

As indicated, this sentence was deleted.

The section headed "Predictors of HBeAg seroconversion at week 48" should be deleted. This is NOT an objective stated in the protocol and is entirely post-hoc.

This was a post-hoc analysis. Although it was not included as an endpoint, it is essential to analyze predictors associated with the endpoint. We moved this section to the supplementary section (S7 Table).

## Table 1

Line 289 should be deleted and the information should be included in the stub.

We deleted the line and added the following text to the table: “Data are presented as mean±SD or number (%).”

The column of p-values should be omitted. It is highly inappropriate to apply formal statistical tests for baseline characteristics in a randomised controlled trial as this is meaningless.

As indicated, the column of p-values was removed.

## Table 2

The information here is better presented as a graph.

As indicated, this table was presented as a Fig. 2.

Line 305 should be deleted and the information should be included in the stub.

We deleted this line and added the following to the table: “Data are presented as mean±SD or number (%).”

Line 313 should be deleted.

As indicated, this line was deleted.

*IMPORTANT* The results of the linear mixed model MUST adjust for the baseline level of the outcome. It is not clear if this is the case.

As indicated, we explained this in the Methods as follows: “The LMM model included repeated measures of numeric variables as dependent variables; group, time, and group×time interaction as fixed effects; baseline outcome as a continuous covariate; and study participant as a random effect.” 

## Table 4

The subheadings are wrong. They must be changed.

We changed the subheadings accordingly. 

## Table 5

Standard deviations should be provided, not standard errors. Alternatively, 

We have provided 95% confidence intervals in place of standard errors.

## Table 6

The footnote explaining that the chi-square test could not be used is wrong. This is because the data presented are wrong. The authors need to provide the number of patients in each group who experienced at least one adverse event. This number cannot exceed the total number of participants. This must be corrected.

This is a very good point. We did not calculate the number, but the incidence. We corrected the Table 6 (now Table 4) accordingly.

# Minor Issues

The authors need to define all abbreviations and acronyms on first use. HBsAg is defined, but HBeAg is not, for example. They need to check the manuscript for similar occurrences.

As indicated, we defined all abbreviations and acronyms on first use.

The demonstrative pronoun in Line 130 is vague. I read it to mean reference 18. Please improve the sentence to reduce confusion.

As indicated, we changed this expression.

The compound adjective in Line 132 is missing a hyphen, as are those in Lines 158-159 and elsewhere. Please review and modify.

As indicated, we revised these texts. 

What is the limit of detection of HBeAg? That is to say, at what titer of HBeAg is a patient considered HBeAg-positive?

The ARCHITECT HBeAg assay detection limit is 0.5 PEI U/mL.

Line 238: replace "numeric" with "continuous"

As indicated, we changed the word.

Line 247: replace "observed" with "observation"

As indicated, we changed the word.

# Recommendation

The clinical trial described in the manuscript contains many unexplained and important deviations from the protocol. I am unable to support the approval of this manuscript for publication in the journal until these issues are corrected.

Reviewer #5: This study by Heo J et al provides important information for how the peg-IFN may benefit on the the HBeAg positive CHB patients who remained HBeAg positive after 5-6 years of NA treatment. This is a well conduct study.

There are few questions remained to be answered before acceptance:

1. When the ALT elevation occurred in the peg-IFN arm? The on-treatment ALT flare is worth to be noted especially when the higher HBeAg seroconversion in the peg-IFN group. Have these patients checked with the autoimmune hepatitis markers to exclude the possibility of IFN related autoimmune issue? It would be great if the ALT kinetics be provided in addition to HBsAg kinetics as well.

We excluded patients with other liver diseases such as autoimmune hepatitis at the time of inclusion and at the time of ALT flare. We also provided HBsAg kinetics and ALT kinetics according to HBeAg seroconversion at week 48 in a S2 Fig From baseline to week 48, HBsAg level was significantly lower in patients who achieved HBeAg seroconversion at week 48 (all p<0.05). ALT tended to be higher in patients who achieved HBeAg seroconversion at week 48, but there was no statistically significant difference. 

2. What's the reason for ALT elevation in the NA treated arm? It is quite surprising that the proportion for ALT > 1X ULN in the NA arn is as high as 32.4%!

Although the number of patients with ALT elevation in the NA group was not small, the ALT elevations were mild and not persistent. The possible causes of a mild ALT elevation in the NA group may use of a cold medication or due to excessive exercise or excessive work.

3. How many proportion of these two arms patients had reach HBsAg quantification level < 100 by the end of the study? How many proportion reach HBsAg rapid decline by the end of the study, which defined as HBsAg reduction > 0.5 log10IU/mL per year? Since these two endpoints may be the sentinal predictor for subsequent HBsAg loss, it is important to know whether the peg-IFN arm have the higher chance for functional cure.

In PegIFN alfa-2a group, 26.7% of patients had an HBsAg reduction of at least 0.5 log10 IU/mL at week-48, but this number was only 2.7% in the NA group. Eight patients in PegIFN alfa-2a group had an HBsAg titer below 100 IU/mL at week-48, but this number was only 2 in the NA group. We provided these data in a S8 Table.

4. What's the outcome of the dropped out patients? It is also interesting to know what happened when these treated 5-6 years patients be discontinued from antiviral therapy.

We do not know the outcomes of all patients who dropped out, because some of them did not return to our study center. Most of them were still receiving nucleos(t)ide therapy because there was no evidence of HBsAg seroclearance.

---

## [Decision Letter · Decision Letter 1]

11 May 2022

PONE-D-22-01751R1Effect of switching from nucleos(t)ide maintenance therapy to PegIFN alfa-2a in patients with HBeAg-positive chronic hepatitis B: A randomized trialPLOS ONE

Dear Dr. Heo,

Thank you for submitting your manuscript to PLOS ONE. After careful consideration, we feel that it has merit but does not fully meet PLOS ONE’s publication criteria as it currently stands. Therefore, we invite you to submit a revised version of the manuscript that addresses the points raised during the review process. ==============================

The comments of the Reviewer 5 have not sufficiently responded and revised. 

We look forward to receiving your revised manuscript.

Kind regards,

Jee-Fu Huang, M.D., Ph.D.

Academic Editor

PLOS ONE

Journal Requirements:

Additional Editor Comments (if provided):

Please respond sufficiently to the comments of the Reviewer 5.

Reviewers' comments:

Reviewer's Responses to Questions

**Comments to the Author**

1. If the authors have adequately addressed your comments raised in a previous round of review and you feel that this manuscript is now acceptable for publication, you may indicate that here to bypass the “Comments to the Author” section, enter your conflict of interest statement in the “Confidential to Editor” section, and submit your "Accept" recommendation.

Reviewer #1: All comments have been addressed

Reviewer #2: All comments have been addressed

Reviewer #3: All comments have been addressed

Reviewer #4: All comments have been addressed

Reviewer #5: (No Response)

2. Is the manuscript technically sound, and do the data support the conclusions?

Reviewer #1: (No Response)

Reviewer #2: Yes

Reviewer #3: Yes

Reviewer #4: (No Response)

Reviewer #5: Partly

3. Has the statistical analysis been performed appropriately and rigorously? 

Reviewer #1: (No Response)

Reviewer #2: Yes

Reviewer #3: Yes

Reviewer #4: (No Response)

Reviewer #5: N/A

4. Have the authors made all data underlying the findings in their manuscript fully available?

Reviewer #1: (No Response)

Reviewer #2: Yes

Reviewer #3: Yes

Reviewer #4: (No Response)

Reviewer #5: Yes

5. Is the manuscript presented in an intelligible fashion and written in standard English?

Reviewer #1: (No Response)

Reviewer #2: Yes

Reviewer #3: Yes

Reviewer #4: (No Response)

Reviewer #5: No

7. PLOS authors have the option to publish the peer review history of their article (what does this mean?). If published, this will include your full peer review and any attached files.

Reviewer #1: No

Reviewer #2: No

Reviewer #3: No

Reviewer #4: No

Reviewer #5: No

6. Review Comments to the Author

Reviewer #5: The authors response make me quite confused than the original drafting.

1. I would suggest to add the limitation in their response to the reviewer into the limitation paragraph.

2. As for their response to the reviewer 1 point 6 and reviewer 2 point 4, I think the discordance of the patients number and the statement makes me really confused about what happened actually to these patients. Would it be possible to make it clear? (Please find a native English speaker to check the fluency before re-submitting the response to the reviewers).

---

## [Author Response · Author response to Decision Letter 1]

8 Jun 2022

As indicated, we have reviewed our reference list and confirm that it is complete and correct.

Additional Editor Comments (if provided):

Please respond sufficiently to the comments of the Reviewer 5.

Reviewer #5: The authors response make me quite confused than the original drafting.

1. I would suggest to add the limitation in their response to the reviewer into the limitation paragraph.

We have added the limitations mentioned in our response to the reviewer to the limitation paragraph in the manuscript.

2. As for their response to the reviewer 1 point 6 and reviewer 2 point 4, I think the discordance of the patients number and the statement makes me really confused about what happened actually to these patients. Would it be possible to make it clear? (Please find a native English speaker to check the fluency before re-submitting the response to the reviewers).

We are very sorry for the confusion. We did not provide sufficient explanation about Table 4 (now Table 2) in our first response to the reviewer. Table 4 (now Table 2) contained information only on cases with HBV DNA elevation that was newly developed at each time point during the study period. The data did not include cases where HBV DNA elevation developed at a previous time point. Therefore, the number of patients with HBV DNA>20,000 IU/mL could be larger than that of patients with HBV DNA>20 IU/mL or >2,000 IU/mL if the rescue antiviral therapy were delayed. We added the above explanation to the footnote of the table and added an additional column to the table to avoid such confusion. That additional column indicated the proportion of patients with HBV DNA level >20 IU/mL, 2,000 IU/mL and 20,000 IU/mL at each time point, excluding cases lost to follow-up or with rescue NA retreatment. This new column shows that the number of patients is more concordant across each HBV DNA level at each time point.

## We also add the answer to previous comment 1 by reviewer #5 

We added data on ALT kinetics during treatment for the PegIFN and NA groups to S2 Fig. Elevated ALT was observed from week 12 to week 48 in the PegIFN group, and the values for this group were significantly higher than those of the NA group (all p<0.001).

---

## [Editor Report · Decision Letter 2]

16 Jun 2022

Effect of switching from nucleos(t)ide maintenance therapy to PegIFN alfa-2a in patients with HBeAg-positive chronic hepatitis B: A randomized trial

PONE-D-22-01751R2

Dear Dr. Heo,

We’re pleased to inform you that your manuscript has been judged scientifically suitable for publication and will be formally accepted for publication once it meets all outstanding technical requirements.

Kind regards,

Jee-Fu Huang, M.D., Ph.D.

Academic Editor

PLOS ONE
---

## [Editor Report · Acceptance letter]

14 Jul 2022

PONE-D-22-01751R2 

Effect of switching from nucleos(t)ide maintenance therapy to PegIFN alfa-2a in patients with HBeAg-positive chronic hepatitis B: A randomized trial 

Dear Dr. Heo:

I'm pleased to inform you that your manuscript has been deemed suitable for publication in PLOS ONE. Congratulations! Your manuscript is now with our production department. 

Kind regards, 

on behalf of

Dr. Jee-Fu Huang 

Academic Editor

PLOS ONE